# Inducible MLL-AF9 Expression Drives an AML Program during Human Pluripotent Stem Cell-Derived Hematopoietic Differentiation

**DOI:** 10.3390/cells12081195

**Published:** 2023-04-20

**Authors:** Branco M. H. Heuts, Saioa Arza-Apalategi, Sinne G. Alkema, Esther Tijchon, Laura Jussen, Saskia M. Bergevoet, Bert A. van der Reijden, Joost H. A. Martens

**Affiliations:** 1Faculty of Science, Department of Molecular Biology, Radboud University, 6525 GA Nijmegen, The Netherlands; bheuts@science.ru.nl (B.M.H.H.);; 2Radboud University Medical Center, Department of Laboratory Medicine, Laboratory of Hematology, 6525 GA Nijmegen, The Netherlands

**Keywords:** MLL-AF9, induced pluripotent stem cells (iPSCs), hematopoiesis, myeloid, differentiation, bioinformatics, oncogene

## Abstract

A t(9;11)(p22;q23) translocation produces the MLL-AF9 fusion protein, which is found in up to 25% of de novo AML cases in children. Despite major advances, obtaining a comprehensive understanding of context-dependent MLL-AF9-mediated gene programs during early hematopoiesis is challenging. Here, we generated a human inducible pluripotent stem cell (hiPSC) model with a doxycycline dose-dependent MLL-AF9 expression. We exploited MLL-AF9 expression as an oncogenic hit to uncover epigenetic and transcriptomic effects on iPSC-derived hematopoietic development and the transformation into (pre-)leukemic states. In doing so, we observed a disruption in early myelomonocytic development. Accordingly, we identified gene profiles that were consistent with primary MLL-AF9 AML and uncovered high-confidence MLL-AF9-associated core genes that are faithfully represented in primary MLL-AF9 AML, including known and presently unknown factors. Using single-cell RNA-sequencing, we identified an increase of CD34 expressing early hematopoietic progenitor-like cell states as well as granulocyte-monocyte progenitor-like cells upon MLL-AF9 activation. Our system allows for careful chemically controlled and stepwise in vitro hiPSC-derived differentiation under serum-free and feeder-free conditions. For a disease that currently lacks effective precision medicine, our system provides a novel entry-point into exploring potential novel targets for personalized therapeutic strategies.

## 1. Introduction

The majority of acute myeloid leukemia (AML) cases are associated with non-random chromosomal translocations that can result in gene rearrangements [1]. Many encode into abnormal transcriptional activators that can alter gene expression necessary for myeloid development, cell proliferation, and/or survival [2]. Consequently, a cell state that is susceptible to leukemic transformation can be established [3]. The potential targeting of these fusion transcripts has become a major focus for the development of novel therapeutics.

Age-related cells of origin and/or additional genetic lesions can lead to changes in the pathology and clinical outcome [4,5,6]. For instance, the MLL-AF9 (KMT2A-MLLT3) fusion, associated with up to 5% of adult AML and in 25% of de novo AML in children [4], initiates transformation in rapidly cycling myeloid progenitors [7]. Moreover, it has been shown that neonatal cells are inherently more susceptible to MLL-AF9-mediated immortalization than adult cells, and co-existing mutations may contribute to the aggressiveness of the disease [5,6].

Despite these advances, obtaining a comprehensive understanding of context-dependent MLL-AF9-mediated gene programs is challenging. Specifically, investigating gene programs during human hematopoiesis poses technical and ethical challenges [8]. The controllable oncogene activation during in vitro differentiation of human-induced pluripotent stem cells (hiPSCs) provides a unique opportunity to dissect the molecular differences that help to drive malignancy at the earliest stages of human hematopoietic development [9]. Not only does such a system allow for the control of pluripotency, early germ layer induction, hemogenic specification, endothelial-to-hematopoietic transition, and control of the microenvironment driving lineage commitment [8], but also control in dosage, timing, and at which differentiation stage the oncofusion is activated.

Here, we developed an MLL-AF9 inducible hiPSC model with which we performed in vitro early myelomonocytic differentiation. We observed immunophenotypical differences during hiPSC-derived hematopoiesis after MLL-AF9 activation. We determined the transcriptional and epigenetic programs mediated by MLL-AF9 as a single oncogenic hit. Specifically, we sought to define high-confidence core genes by comparing the identified MLL-AF9-mediated gene programs to primary MLL-AF9 AML samples. Consequently, we exploited these high-confidence core genes to identify a context-dependent response to MLL-AF9 expression at various levels of blood cell differentiation.

## 2. Materials and Methods

### 2.1. iMLL-AF9 Generation

The doxycycline inducible MLL-AF9 construct was inserted using an AAVS1 homology donor vector and CRISPR-Cas9 into MML6838.Cl2 (MIP 7 Cl.1-2) (Sanquin, Amsterdam, The Netherlands) cell line [10], as described by Mandoli et al. [11]. In short, iPSCs were nucleofected with a donor vector containing an inducible promoter that activates expression of the cloned gene and a gene targeting vector for the AAVS1 locus. Transfected cells were cultured in Vitronectin (Life Technologies, Carlsbad, CA, USA)-coated 6-well plates in E8 media (Life Technologies) and supplemented with 10 μM of rock inhibitor for 24 h. Cells were dissociated using accutase (Life Technologies) and selected with 0.25 μg/mL puromyocin for 14 days. Positive clones were selected via PCR using MLL-AF9 fusion primers:
-MLL-AF9_fwd_PCR: GGACTTTTCACTTCAAGAATCTTTTCTTTTGG;-MLL-AF9_rev_PCR: CCAAAAGAAAAGATTCTTGAAGTGAAAAGTCC.

### 2.2. Cell Culture

Transfected iPS cells were cultured in mTeSR™ Plus (STEMCELL™ Technologies, Köln, Germany)-coated 6-well plates at 37 °C and supplemented with 1% Penicillin/Streptomycin on Vitronectin XF™ (STEMCELL™ Technologies). After thawing, cells were treated with 10 μM of rock inhibitor for 24 h and routinely split every 3–4 days on freshly coated plates using ReLeSR (STEMCELL™ Technologies).

### 2.3. Myelomonocytic Differentiation

For differentiating iPSCs to HPCs expressing CD34, CD45, and CD43, a STEMdiff™ Hematopoietic Kit (STEMCELL™ Technologies) was used. Until day 0, the cells were cultured in mTeSR™ Plus (STEMCELL™ Technologies)-coated 6-well plates at 37 °C and supplemented with 1% Penicillin/Streptomycin on Vitronectin XF™ (STEMCELL™ Technologies). On day 6 and routinely every other day afterward (day 8, 10, 12, etc.), doxycycline (16 ng/mL) was supplemented to induce MLL-AF9 expression. We chose day 6 to start doxycycline treatment because, at this stage, HPCs are most likely not (partially) differentiated. We estimated that the first HPCs would be formed in the hematopoietic clusters around day 8. Therefore, day 6 provided the largest window of opportunity to study HPC transformation into (pre-)leukemic states. As a consequence, the MLL-AF9 fusion could target downstream genes that are important during early hematopoiesis. After 12 days, the cells residing in the supernatant were transferred to 6-well plates and the medium was replaced by Stemline^®^ II Hematopoietic Stem Cell Expansion Medium (Sigma-Aldrich, Saint Louis, MO, USA) supplemented with 1% Penicillin/Streptomycin, 1:100 insulin-transferrin-selenium-ethanolamine (ITS-X) (Thermo Fisher Scientific, Waltham, MA, USA) and cytokines (50 ng/mL IL-3, 50 ng/mL FLT3-L, 50 ng/mL SCF, 50 ng/mL M-CSF, 10 ng/mL TPO) (Miltenyi Biotec B.V., Leiden, The Netherlands) to induce the monocyte differentiation. The medium was refreshed every 2–3 days, and MLL-AF9 expressing cells were kept continuously in doxycycline upon further analysis or exhaustion.

### 2.4. RNA Extraction and Real-Time PCR

RNA was isolated from 1E5 cells using Quick-RNA™ Microprep (Zymo Research, Breisgau, Germany) and reverse transcribed using an iScript™ cDNA Synthesis Kit (Bio-Rad, Hercules, CA, USA). Real-time amplification was performed using iQ SYBR Green mix (Bio-Rad) on a CFX96™ Real-Time System (Bio-Rad) and quantified using Bio-Rad CFX Manager. To calculate the relative expression levels, Glyceraldehyde 3-phosphate dehydrogenase (GAPDH) was used. We used the following primers:
-MLL-AF9_fwd_qPCR: CGA AGA CGA AGA CGA GGC GG;-MLL-AF9_rev_qPCR: AGA CAC ATT CTG CAG CAG ATC GTG;-AAVS1_fwd: CAG TTA CAT TGG ATC CCT GCA GGC.

### 2.5. Western Blot

The iPSCs were cultured in E8 media (Life Technologies, Carlsbad, CA, USA) induced with 50 ng/mL doxycycline for 24 h. As a control, we used untreated iPSCs.

A 7% resolving gel (11.4 mL 1.25 M Bis-Tris (pH 6.5), 9.34 mL acrylamide (ProtoGel 30%) (National Diagnostics, Atlanta, GA, USA), 400 µL 10% ammonium persulfate (Sigma-Aldrich, Saint Louis, MO, USA), 20 µL TEMED (AMRESCO, Radnor, PA, USA), 19.06 mL MilliQ) and a stacking gel (4.3 mL 1.25 Bis-Tris (pH 6.5), 2 mL acrylamide (ProtoGel 30%) (National Diagnostics), 150 µL 10% ammonium persulfate (Sigma-Aldrich), and 20 µL Temed (AMRESCO)) were used.

Cell pellets were lysed in 60 µL Leammli buffer (4 mL 10% SDS, 1.2 mL 1 M Tris-Cl (pH 6.8), 200 µL 1% bromophenol blue, 2.6 mL H_2_O, 2 mL fresh 1 M DTT) per 1 million cells. Resuspended cells were boiled for 5 min and centrifuged at 4 °C for 10 min at maximum speed. Subsequently, 50 µL of supernatant was loaded for electrophoresis. Proteins were electrophorized in MOPS 1× with 1 g sodium bisulfite (ACROS organics, Geel, Belgium). Proteins were transferred from gel to nitrocellulose in CAPS buffer (pH 11) (Sigma-Aldrich) for 3 h at 400 mA. After transfer, nitrocellulose was blocked with 15 mL milk (1g Skim Milk Powder (Sigma-Aldrich) solved in 50 mL DPBS-T) for 1 h. After blocking, nitrocellulose was washed 3 times with DPBS-T for 10 min. Proteins on nitrocellulose were incubated with 1:5000 primary antibody to MLL-N (Sigma-Aldrich, 05-764) and 1:10,000 primary antibody to vinculin (abcam, ab155120) in 15 mL milk overnight. After incubation, nitrocellulose was washed 3 times with DPBS-T for 10 min. The MLL-N primary antibody was targeted by 1:5000 ratio of secondary donkey antibody to mouse (IRDye^®^ 800 CW) (LI-COR Biosciences, Lincoln, NE, USA) in 15 mL milk. The primary antibody to vinculin was targeted by 1:5000 ratio of secondary donkey antibody to rabbit (IRDye^®^ 680 RD) (LI-COR Biosciences) in 15 mL milk. After incubation, nitrocellulose was washed 3 times with DPBS-T for 10 min.

### 2.6. Flow Cytometric Analysis

On day 15, cells were stained with a monocyte (CD14, CD64) antibody cocktail and kept in the dark for 30 min at room temperature. Samples were then washed 3 times with PBS + 1% BSA. After staining, cells were analyzed on a Beckman Coulter Gallios 10-color (Beckman Coulter, Brea, CA, USA) with the Kaluza G software (2014).

### 2.7. Publicly Available Datasets

Publicly available primary AML RNA-seq data were used from the BeatAML adult cohort and TARGET AML pediatric cohort [12]. The TARGET AML data were generated by the Therapeutically Applicable Research to Generate Effective Treatments (TARGET) initiative, phs000218, managed by the NCI. The GDC TARGET-AML data used for this analysis are available at https://xenabrowser.net/datapages/?cohort=GDC%20TARGET-AML&removeHub=https%3A%2F%2Fxena.treehouse.gi.ucsc.edu%3A443, accessed on 27 January 2021. Information about TARGET can be found at http://ocg.cancer.gov/programs/target (accessed on 27 January 2021). The BeatAML data used for this analysis are available at http://vizome.org/additional_figures_BeatAML.html (accessed on 27 January 2021). Additionally, publicly available ChIP-seq occupancy changes of MLL-AF9 (HA) in MLL-AF9-HA-FKPB12-transformed human cells (HCB1) were obtained from the gene expression omnibus (GSE173599) [13] and visualized in a UCSC browser session (https://genome.ucsc.edu/s/bheuts/hg38_Armstrong_MLLAF9_degradation, accessed on 27 January 2021).

### 2.8. RNA Sequencing

Cells were collected at various timepoints between day 20 and day 28. RNA was isolated from 1E6 cells using Quick-RNA™ Microprep (Zymo Research, Breisgau, Germany) and on-column DNaseI treatment. Library generation was performed on 100 ng RNA using KAPA RNA HyperPrep Kit with RiboErase (HMR) (Kapa Biosystems, Potters Bar, UK), with an RNA fragmentation of approximately 300 bp fragments for 6 min at 94 °C. Library size distribution was measured using High Sensitivity DNA analysis (Agilent, Santa Clara, CA, USA) on an Agilent 2100 Bioanalyzer and its corresponding software (version B.02.08.SI648). Libraries with average sizes between approximately 300–400 bp were used for sequencing via a NextSeq 500 system (Illumina, San Diego, CA, USA).

The fastq files were mapped to the reference human genome hg38 through the Seq2science pipeline (https://github.com/vanheeringen-lab/seq2science, accessed on 27 January 2021) (STAR as default aligner). Count normalization and differential expression analysis (1.5 fold change, *p*-value < 0.05) was performed using DESeq2 in R [14]. Batch correction was performed with Limma’s removeBatchEffect [15]. Subsequently, enriched gene sets were determined via Gene Set Enrichment Analysis (GSEA) [16], and GO-term enrichment was performed using GAGE [17].

### 2.9. Assay for Transposase-Accessible Chromatin Using Sequencing

Cells were harvested on day 23 or day 24 from two independent differentiation experiments. Suspension cells (50,000 per sample) were washed twice in ice cold PBS. The pellet was resuspended in 1:1 ice cold PBS and 2× lysis buffer (1 M Tris/HCl pH 7.5, 5 M NaCl, 0.5 M MgCl_2_, 10% NP40). The cells were then centrifuged (300× *g*) for 30 min at 4 °C, and the pellet was resuspended with 24 µL clean-up buffer (5 M NaCl, 0.5 M EDTA, 10 mg/mL Proteinase K, 10% SDS) whilst being kept on ice. Then, 1 µL Tn5 was added to each sample for tagmentation. The nuclei were heated for 6 min at 37 °C with 650 rpm agitation. Immediately after incubation, 9 µL clean-up buffer was added to each sample and incubated for 30 min at 40 °C with 650 rpm agitation. The samples were purified with a normal phase 2× SPRI purification and amplified by PCR using Nextera primers (Illumina, San Diego, CA, USA) and KAPA HiFi HotStart ReadyMix (Roche). Tagmented DNA was amplified using a five-cycle PCR protocol: 5 min at 72 °C; 45 s at 98 °C; 5× cycles: {15 s at 98 °C; 30 s at 63 °C; 30 s at 72 °C}; 60 s at 72 °C; hold at 12 °C. A reverse phase 0.65× SPRI bead purification step was performed, followed by a 1.5× SPRI beads clean-up. Next, a second PCR amplification program was used to further amplify the tagmented DNA. The program was identical to the first PCR program, including the Nextera primers, except for the number of cycles, which was determined by qPCR. Real-time amplification was performed using iQ SYBR Green mix (Bio-Rad, Hercules, CA, USA) on a CFX96™ Real-Time System (Bio-Rad) and quantified using Bio-Rad CFX Manager. The following program was used: 45 s at 98 °C; 40× cycles: {15 s at 98 °C; 30 s at 63 °C; 30 s at 72 °C}; Melt curve: {0.05 at 65 °C; 0.5 at 95 °C}. The Ct value + 2 was used as PCR cycles for the second PCR amplification step. Subsequently, two successive 1.5× SPRI clean-ups were performed to generate the ATAC libraries. The DNA was stored at −20 °C, and an aliquot was used to measure the DNA concentration with a DeNovix dsDNA HS Assay Kits (Life Technologies, Carlsbad, CA, USA) on a DeNovix Spectrophotometer/Fluorometer (DS-11) (DeNovix, Wilmington, UK). Library size distribution was measured using High Sensitivity DNA analysis (Agilent, Santa Clara, CA, USA) on an Agilent 2100 Bioanalyzer and its corresponding software. DNA fragment distribution followed a mononucleosomal-like pattern at approximately 200 bp and 320 bp. The libraries were sequenced using a NextSeq 500 system (Illumina, San Diego, CA, USA).

Trimming, aligning, and peak calling was performed via Seq2science (https://github.com/vanheeringen-lab/seq2science, accessed on 27 January 2021) (Trimmer: fastp, aligner: bwa-mem2, peak caller: macs2).

### 2.10. ANalysis Algorithm for Networks Specified by Enhancers (ANANSE)

MLL-AF9-specific transcription factors were predicted using ANANSE v0.4.0 [18]. TF binding profiles were predicted using narrow peaks and sequence alignment data (BAM) from ATAC-seq, motif scores, and average ReMap ChIP-seq coverage, as described previously by Xu et al. [18]. Subsequently, gene regulatory networks (GRNs) were determined using our normalized (TPM) RNA-seq results. For the TF prediction, we used human genome assembly hg38. To obtain key TFs, we only show TFs that were predicted to be important in two independent analyses.

### 2.11. Peak Calling

Trimming, aligning, and peak calling was performed via Seq2science (https://github.com/vanheeringen-lab/seq2science, accessed on 27 January 2021) (Trimmer: fastp, aligner: bwa-mem2, peak caller: macs2). Peaks were annotated using ChIPpeakAnno and GREAT [19,20].

### 2.12. Comparative Bulk RNA Sequencing Analysis

Systematically statistical testing was performed using ggsignif [21]. Next, UMAP and unsupervised hierarchical clustering was performed using umap (https://github.com/tkonopka/umap, accessed on 27 January 2021) and dendextend [22].

### 2.13. Single-Cell RNA Sequencing

On day 23, cells from each condition were sorted into separate 384-well plates with a Becton Dickinson Aria Flow cytometry sorter (Becton, Dickinson and Company, Franklin Lakes, NJ, USA), using a 100 µM nozzle. After this, plates were frozen at −80 °C and thawed before further processing. Sequencing library generation was performed according to the RAID-seq protocols [23] with adaptations; libraries were generated without immunostaining and Barcode Compensation Primers as we only sequenced mRNA transcripts. Thanks to the use of an Agilent 2100 Bioanalyzer and High Sensitivity DNA analysis (Agilent, Santa Clara, CA, USA), DNA fragment sizes were determined. Libraries with average sizes between approximately 300–400 bp were used for sequencing via a NextSeq 500 system (Illumina, San Diego, CA, USA).

The sequenced libraries were then processed into spliced and unspliced count matrices using Seq2science (https://github.com/vanheeringen-lab/seq2science, accessed on 27 January 2021) (CELSeq2 workflow; RNA velocity/Kallisto). Then, quality control metrics were determined using a custom scRNA-seq pre-processing workflow (https://github.com/Rebecza/scRNA-seq, accessed on 27 January 2021), and downstream analyses, such as cell cycle scoring and differential expression analysis, were computed using the scRNA-seq integration workflow with the SCTranform normalization method of the Seurat package in R [24]. The number of dimensions were determined using an elbow plot; 13 dimensions were chosen (Appendix A). We used the default clustering resolution (Appendix A).

Identifying cross-dataset matched biological states (anchors) can be challenging. For the pre-subset dataset, we revealed generally lower anchor scores for cluster 0 compared to other clusters, implying subpar integration for these cells (Appendix A). Therefore, profound biological understanding may be compromised for cluster 0. On the contrary, other anchors successfully recovered matching cell states, even with MLL-AF9-induced dataset differences. Therefore, we removed cells from cluster 0 and performed the quality assessment and downstream analyses with the post-subset dataset. We set the dimensions to 14 (Appendix A) and kept the clustering resolution at default parameters. We applied a zero-preserving imputation strategy using ALRA [25].

## 3. Results

### 3.1. Generation of an Inducible MLL-AF9 Human Pluripotent Stem Cell Model

To investigate the molecular mechanisms mediated by MLL-AF9 expression during human hematopoiesis, we established a doxycycline inducible human MLL-AF9 iPSC model. An MLL-AF9 inducible iPSC clone derived from human megakaryoblasts was similarly generated, as previously described [10,11]. A dose-dependent expression of MLL-AF9 in iPSCs was determined via qPCR to match the physiological MLL-AF9 expression levels in the MLL-AF9-positive human leukemia monocytic cell line THP-1 (Figure 1A). MLL-AF9 expression reached physiological levels of THP-1 between 14 and 50 ng/mL doxycycline. Indeed, it was confirmed that MLL-AF9 protein expression used Western blotting after doxycycline induction (Figure 1B). For this, we used an antibody that recognizes the N-terminal region of MLL, resulting in the detection of endogenous wild-type MLL (variants) and the MLL-AF9 fusion protein. These results demonstrated that doxycycline-induced promoter activation causes MLL-AF9 expression on the transcriptome and proteome level.

Next, to evaluate whether induced MLL-AF9 expression deregulates differentiation along the early progenitor to the myeloid axis, we induced iPSC-derived hematopoiesis (Figure 1C). First, iPSCs were differentiated into a mesoderm-like state, a prerequisite for the specification of hemogenic endothelial cells [8]. Subsequently, early hematopoietic progenitor cells (HPCs) were generated and transferred to initiate myelomonocytic differentiation using a monocytic cytokine cocktail. Depending on the doxycycline treatment, we either generated iMonocytes or iMLL-AF9 cells. We did not perform any subsequent sorting strategy to isolate the iMonocytes from the myeloid population. To determine the cell expansion capacity of iMonocytes and iMLL-AF9 cells, we calculated the total cumulative cell count (per mL) for each condition (Figure 1D). Up to day 22 of the myelomonocytic differentiation cell expansion was comparable for each condition. Thereafter, iMLL-AF9 cell expansion increased and continued until day 66 before exhaustion occurred, whereas iMonocytes were exhausted after day 30. These results indicated that the induction of MLL-AF9 deregulates normal hiPSC-derived hematopoiesis. To identify immunophenotypic alterations upon MLL-AF9 activation, we measured the cell-surface marker expression for monocytes (CD14 and CD64) using flow cytometry (Figure 1E). At day 15 of the myelomonocytic differentiation, flow cytometry analysis revealed a CD14- and CD64-enriched cell population for iMonocytes, whereas iMLL-AF9 cells were almost entirely devoid of these monocytic markers. In addition, to confirm persisted HPC gene marker expression upon MLL-AF9 induction, we compared gene expression levels of key HPC gene markers (*CD34* and *SPN*) between iMonocytes and iMLL-AF9 cells after day 20 of the differentiation protocol (Figure 1F and Appendix A). A significantly higher expression of *CD34* and *SPN* implies HPC gene programs persisted as a consequence of MLL-AF9 expression.

Together, these results suggest that induced MLL-AF9 expression deregulates normal myelomonocytic differentiation, consequently generating cells with higher cell expansion potential. Moreover, without doxycycline induction, these genetically modified iPSCs were still viable for myelomonocytic differentiation, demonstrating the potential of this doxycycline-dependent MLL-AF9 activation model.

### 3.2. MLL-AF9-Induced Early Hematopoietic Progenitor Cells Express Cord Blood and Leukemia Related Genes

To identify genes that could be relevant for MLL-AF9 AML development during early hematopoiesis, we sequenced RNA from iMonocytes and iMLL-AF9 cells. Principal Component Analysis (PCA) revealed that the largest fraction of variation, PC1, could be accounted to MLL-AF9-induced expression changes (Figure 2A and Appendix A). Differentially expressed genes were determined to examine the genes that were significantly affected by MLL-AF9 expression, revealing 379 upregulated genes (*p*-value < 0.05, log2 fold change > 1.5) and 1080 downregulated genes (*p*-value < 0.05, log2 fold change < −1.5) (Figure 2B and Appendix A). Indeed, we identified a significantly higher expression of critical MLL-AF9 target genes such as *MEIS1*, *HOXA9*, and *HOXA10* (Figure 2B and Appendix A) [26,27]. In addition, the key markers for deregulated myeloid differentiation associated with MLL-AF9 AML were significantly upregulated (Figure 2B and Appendix A), such as *ZNF521* and *CDK6* [28,29]. Genes that translate into monocyte and neutrophil cell-surface markers were downregulated after MLL-AF9 induction, e.g., *CD14*, *FCGR3B*, and *CEACAM3* [30,31,32]. These results suggest that MLL-AF9 activation disrupts gene programs that mediate myelomonocytic differentiation and induced gene expression relevant to MLL-AF9 AML pathogenesis.

To elucidate the underlying biological processes associated with this MLL-AF9-mediated gene response, we performed Gene Ontology (GO) term enrichment analysis. Our iMonocytes were significantly enriched for “leukocyte mediated immunity”, “leukocyte proliferation”, and “MHC protein complex assembly”, signifying white blood cell development and immunity, whereas iMLL-AF9 cells were significantly enriched for biological processes such as “chromosome segregation”, “regulation of cell cycle phase transition”, and “RNA localization” (Figure 2C)—processes that are generally associated with a proliferative phenotype. Overall, these results are in line with our previous flow cytometry results, implying a disturbance of iPSC-derived hematopoiesis upon MLL-AF9 expression.

In order to demonstrate that our iPSC-derived hematopoietic cells are relevant to early hematopoietic development and leukemia gene profiles, we performed Gene Set Enrichment Analysis (GSEA). For this, we used multiple curated gene sets for neonatal HSC signatures identified by Jaatinen et al. [33], Novershtern et al. [34], and Eppert et al. [35] (Figure 2D, left column). Indeed, our iMLL-AF9 model correlated with HSC signatures, as we observed significant enrichment scores resulting from our upregulated iMLL-AF9 genes. Moreover, MLL-AF9-induced CD34^+^ cord blood cell signatures by Horton et al. [6], leukemic stem cells (LSCs) by Eppert et al. [35], signatures for cell cycle progression by Kanehisa et al. [36], and an AML signature by Köhler et al. [37] were enriched in our iMLL-AF9 cells (Figure 2D, right column). Our iMonocytes were inversely correlated with HSCs, LSCs, and MLL-AF9 expressing CD34^+^ cord blood signatures (Figure 2D). Together, we demonstrated significant upregulation of key MLL-AF9 AML marker genes after MLL-AF9 induction as a single oncogenic hit. Furthermore, these results provide a strong indication that our iMLL-AF9 cells share a transcriptional profile with early myeloid progenitors and are relevant to AML-like overproduction of such early myeloid progenitors.

### 3.3. Identifying Key Transcription Factors Driving MLL-AF9-Induced Early Hematopoietic Progenitor Cells

Transcription factors (TFs) are important regulators during the progression of normal hematopoiesis [38]. In MLL-AF9 AML, numerous TFs are considered to be oncogenic and/or important for leukemogenesis and/or maintenance [39]. To identify important TFs mediated by MLL-AF9 expression, we used a network-based TF prediction method that prioritizes TFs based on transcriptomic and chromatin accessibility data [18]. For this, we performed RNA-seq and Assay for Transposase-Accessible Chromatin using sequencing (ATAC-seq) with the use of iMonocytes and iMLL-AF9 cells (Appendix A). First, genome-wide TF binding profiles were predicted using three types of input features: signal intensity at chromatin accessible regions, TF binding motifs scores, and the average ReMap ChIP-seq coverage [40]. Subsequently, differential gene regulatory networks were determined using gene expression information, resulting in an inferred ‘influence score’—a measure of importance to which a TF can explain the transcriptional differences between two cell states. In total, 12 TFs were predicted to be important in MLL-AF9 expressing HPCs (Figure 2E). To determine MLL-AF9 DNA binding at the loci of our predicted TFs, we analyzed publicly available MLL-AF9 ChIP-seq data from genetically modified human cord blood cells and identified nine as direct MLL-AF9 target genes (Appendix A) [27]. Amongst these was *PBX3*, an important TF in MLL-AF9 expressed cells, which has previously been shown to be important in leukemogenesis and maintenance, as well as a direct target of MLL-AF9 [13,41,42]. Interestingly, GATA2 and SP8 were predicted as the most influential TFs in iMLL-AF9 cells, yet their roles in this context remain elusive. Together, these results reveal an instructive set of transcription factors that likely govern MLL-AF9-induced gene programs. The majority of the predicted TFs are indeed direct MLL-AF9 targets, strongly implying that the novel factors could be involved in MLL-AF9-mediated leukemogenesis and/or maintenance.

### 3.4. Comparative Gene Expression Profiling Reveals MLL-AF9-Associated Core Genes in Primary AML

To identify MLL-AF9-specific genes that are faithfully represented in primary AML, we compared our iMLL-AF9 model to MLL-AF9-positive primary AMLs. First, we used our iMLL-AF9 model to uncover MLL-AF9-associated genes (Figure 3A). For each MLL-AF9-associated gene, expression levels were statistically tested by comparing MLL-AF9-positive primary AML to other AML subtypes. For this, we used transcriptomic data from the TARGET AML cohort (pediatric) and BeatAML (adult) cohort [12]. We identified 61 core genes for each cohort: 39 high-confidence genes that were significantly upregulated in both AML cohorts and 22 unique genes that were upregulated in either one (Figure 3A,B, Appendix A). In agreement with the previous results (Figure 2B), we identified known MLL-AF9 targets, such as *HOXA10* and *ZNF521* [28,43]. Based on this analysis, we were able to uncover MLL-AF9-associated core genes that are faithfully represented in primary AML.

Furthermore, to uncover the specificity of the core genes for MLL-AF9 (KMT2A-MLLT3) leukemia, we examined their expression levels in various AML subsets (Figure 3C). Indeed, a higher expression of these core genes correlated well with MLL-AF9 AMLs. Moreover, to determine the discriminative power of MLL-AF9-associated core genes in leukemia subclassification, we performed uniform manifold approximation and projection (UMAP) (Figure 3D,E), as well as hierarchical clustering (Appendix A). In both the pediatric (TARGET AML) and the adult cohort (BeatAML), we were able to define clusters of MLL-AF9 AML patients. Interestingly, the dynamic gene expression within the core genes also allowed for the clustering of other cytogenetic abnormalities, such as RUNX1-RUNX1T1, CBFB-MYH11, or AML patients with a CEBPA mutation. Altogether, these results suggest that we were able to effectively classify primary MLL-AF9 and other AMLs using these core genes.

Next, to determine which of the core genes are directly regulated by the MLL-AF9 fusion protein, we included MLL-AF9 DNA binding from publicly available MLL-AF9-transduced CD34^+^ cord blood data in our analysis [13]. In total, 18 genes were identified as direct MLL-AF9 targets (Appendix A), including known (e.g., *HOXA10*) and novel (e.g., *SKIDA1*) targets. Indeed, all 18 genes were identified as high-confidence MLL-AF9-associated genes that were significantly upregulated in both AML cohorts. In summary, these results suggest that we were able to uncover high-confidence MLL-AF9-associated core genes that are faithfully represented in primary AML. Our syngeneic model served as an important entry point in uncovering these known and presently unknown markers.

### 3.5. MLL-AF9 Expression Affects Distinct Cellular Populations during Early Hematopoietic Development

To determine the effects of MLL-AF9 at various levels of blood cell differentiation, we performed single-cell RNA sequencing using 265 iMonocytes and 774 iMLL-AF9 cells. First, we aligned cross-dataset pairs of cell identities based on a shared set of variable genes and performed UMAP nonlinear dimensionality reduction as well as Louvain clustering to visualize and explore our data. This revealed multiple integrated cell populations between our two conditions (Figure 4A). We defined hematopoietic populations based on the expression of established markers (e.g., *CD34*, *CD14*, *GP1BA*, *ITGA2B*, *SIGLEC8*, *HLA-DRA*, *PRG2*), which revealed that the generated blood cells predominantly derived from the myeloid compartment, such as megakaryocyte-, HPC-, granulocyte-monocyte progenitor (GMP)-, basophil-, eosinophil-, (pro-)monocyte-, and dendritic cell (DC)-like cells (Figure 4B and Appendix A). Our data integration revealed a shared set of cell states for which we could determine MLL-AF9-induced gene expression changes.

MLL-AF9 helps to perpetuate ongoing gene expression programs across multiple cellular states [7], thus creating a permissive environment for competitive cell states with preserved cell cycle behavior. To assess whether cell cycle signatures were upregulated after MLL-AF9 induction, we classified each cell in either G1-, G2M-, or S-phase based on canonical cell cycle marker expression (Figure 4C). When compared to iMonocytes, this revealed an increased expression of both S- and G2M-phase genes for most iMLL-AF9 cell clusters. In agreement with our previous cell expansion results (Figure 1D), these data suggest that MLL-AF9 induction affects the DNA replication gene program in most cell states. Thus, this implies that MLL-AF9 expression maintains a cell cycle program in multiple blood cell compartments.

To uncover whether certain cell populations expanded more in response to MLL-AF9 expression, we normalized for the cell count in iMonocytes and iMLL-AF9 cells (Figure 4D and Appendix A). It was revealed that both cluster 1 (annotated as CD34^+^ HPCs) and cluster 2 (annotated as GMPs) were more expanded after MLL-AF9 induction, which is in concordance with MLL-AF9 studies in human neonatal cells [6]. In addition, cluster 6 (annotated as DCs) became more abundant when MLL-AF9 was induced, which contained cells that expressed pro-angio-/vasculogenic factors (Appendix A). Previous studies identified immature DCs as bipotent cells with vasculogenic potential that could transdifferentiate into endothelial cells in response to high concentrations of VEGF [44]. Whether MLL-AF9 plays a role in the endothelialisation of these DC-like cells needs to be determined in future studies. Furthermore, we observed a decreased (pro-)monocytic-like compartment (cluster 5)—even though these cells were enriched for cell cycle genes—and a non-expanding megakaryocytic-like cell compartment (cluster 0). Taken together, these results suggest that MLL-AF9 affects cell expansion in a cell state-dependent manner.

Next, in order to investigate whether the leukemic potential of iMLL-AF9 stems from specific cell type populations, we examined the expression of our earlier defined MLL-AF9-specific core genes in our identified blood cell populations. In keeping with our prior findings, we observed a higher expression of the core genes in iMLL-AF9 cells compared to iMonocytes (Figure 4E). For iMLL-AF9 cells, the highest expression was observed in clusters 2 (annotated as GMPs), 3, and 4, while other clusters exhibited a lower expression, although overall, their expression was greater compared to the iMonocytes. While the expression of MLL-AF9 core genes was absent in clusters 5 (annotated as (pro-)monocytic), MLL-AF9-independent expression of the core genes was observed in other clusters. Together, this indicates that the effects of MLL-AF9 on core gene expression are cell type-dependent.

Overall, these results reveal a cell state-dependent response to MLL-AF9 expression. MLL-AF9 allows for a competitive environment in which progenitor cell types are more susceptible to transformation. Our model enables the expansion of immature cells with a disturbed myelomonocytic differentiation axis. Thus, this provides a valuable platform to study the MLL-AF9-induced and context-dependent molecular mechanisms of diseases.

## 4. Discussion

In this study, we have established a new platform to examine the consequences of MLL-AF9 expression during human iPSC-derived hematopoietic differentiation. Our syngeneic model can serve as a new entry point to study the MLL-AF9-induced and context-dependent molecular mechanisms of diseases.

The MLL-AF9 fusion protein induces cell expansion and impairs differentiation [6,7]. In agreement with this, we confirmed the deregulation of myelomonocytic differentiation in conjunction with an increase in cell expansion capacity upon MLL-AF9 expression. Indeed, typical gene expression profiles that have well-described roles in both the induction and maintenance of MLL-AF9 AML were enriched following MLL-AF9 activation (e.g., *MEIS1*, *HOXA9*, *HOXA10*, *CDK6*, *ZNF521*) [26,27,28,29,43]. Moreover, we identified MLL-AF9-associated core genes that are faithfully represented in primary MLL-AF9 AML, including known targets such as *HOXA10*, *CDK6*, and *ZNF521*, as well as targets that have a presently unknown role in MLL-AF9 AML, e.g., *SKIDA1*, which seems to be a promising predictor of MLL-rearranged AML [45]. This indicates that we have uncovered a repertoire of gene profiles that includes factors important for leukemic transformation and maintenance in MLL-AF9 AML.

Previously, it has been shown that MLL-AF9 expression sustains already existing gene expression programs in multiple blood cell types [7]. As we can generate various blood cell states along the myeloid differentiation axis with our iPSC model (e.g., megakaryocyte-, granulocyte-, monocyte-like cell states), we assessed the consequences of MLL-AF9 expression in these myeloid blood cell compartments. We revealed a cell state-dependent response to MLL-AF9 expression. In concordance with others [6,7], CD34^+^ HPCs and GMP-like cells were observed to have an increased proliferative ability. Our findings shed light on the cell state-dependent response to MLL-AF9 activation, suggesting several areas for future investigation.

While we did observe the expansion of HPC and GMP-like compartments, it is unclear if this heterogeneity is a consequence of cell state-dependent differential in MLL-AF9 expression levels or the cellular context in which MLL-AF9 expression occurs [46,47]. Additionally, we observed signs of exhaustion after MLL-AF9 induction. This suggests that our cells were not yet immortalized. In this model, the oncogenic hit causes a competitive advantage for cells that have an enriched cell division program, likely priming the cells for malignant transformation. Expressing MLL-AF9 has been shown to be sufficient to induce malignancy in animal models [48,49], whether this is the case with our model needs to be determined.

Importantly, our model allows for excellent control in MLL-AF9 expression activation during iPSC-derived hematopoiesis. Not only are we able to control the cell types in which MLL-AF9 becomes expressed, but we can also control its timing and dosage during the various developmental stages. Furthermore, our iPSCs were differentiated under serum-free and feeder-free conditions, allowing for careful, chemically controlled, and stepwise differentiation.

In summary, we demonstrated that, as an oncogenic hit, inducing MLL-AF9 expression disturbs early human iPSC-derived hematopoiesis in a cell state-dependent manner. By exploiting this system, we can provide a source of biomarkers that are faithfully represented in primary MLL-AF9 AML, including factors that have a presently unknown role in MLL-AF9-mediated programs. As this disease currently lacks effective precision medicine, finding potential novel targets lays the foundation for personalized therapeutic strategies.

## Figures and Tables

**Figure 1 cells-12-01195-f001:**
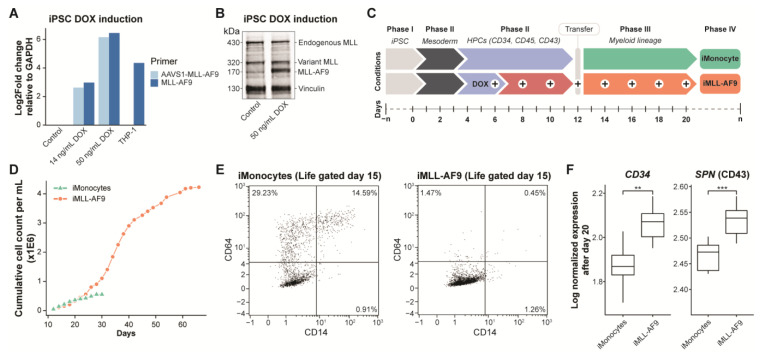
Abnormal human iPSC-derived differentiation along the early hematopoietic monocyte axis. (**A**) RT-qPCR analysis of iPSCs before and after induction of MLL-AF9 with 14 or 50 ng/mL doxycycline for 24 h. THP-1, an MLL-AF9-positive cell line, was used as a positive control. The colors represent two sets of primer pairs. AAVS1-MLL-AF9 represents the combination of a primer pair on the AAVS1 locus and MLL-AF9, while MLL-AF9 represents a primer pair covering the fusion point of MLL-AF9. GAPDH was used as an endogenous control. (**B**) Western blot analysis identified MLL-AF9 protein expression in iPSCs after 24 h stimulation with 50 ng/mL doxycycline using an antibody targeting MLL-N, resulting in the detection of endogenous MLL and the MLL-AF9 fusion protein. Vinculin (130 kDa) was used a loading control. (**C**) A schematic representation of iPSCs to iMonocyte or iMLL-AF9 differentiation, highlighting the induced changes along the differentiation axis. The plus signs signify the induction/maintenance of doxycycline at the respective days. “Transfer” indicates the transfer of cells in the supernatant to a new culture plate supplemented with Stemline II medium and monocyte cytokine cocktail. Cells were maintained until exhaustion occurred. (**D**) Total cumulative expansion of cells grown (per mL) during monocytic differentiation for each condition. (**E**) At day 15 of the differentiation protocol, flow cytometric analysis was performed using a cocktail of antibodies that target monocyte cell-surface markers (CD64 and CD14) on iMonocytes and iMLL-AF9 cells. (**F**) Boxplots depicting the mean log-normalized expression values of *CD34* and *SPN* (CD43) for iMonocytes versus iMLL-AF9. The whiskers represent standard deviation; edges depict the inter-quartile ranges. The black center line represents the median. Two and three asterisks signify *p*-value < 0.01 and *p*-value < 0.001, respectively (one-way ANOVA).

**Figure 2 cells-12-01195-f002:**
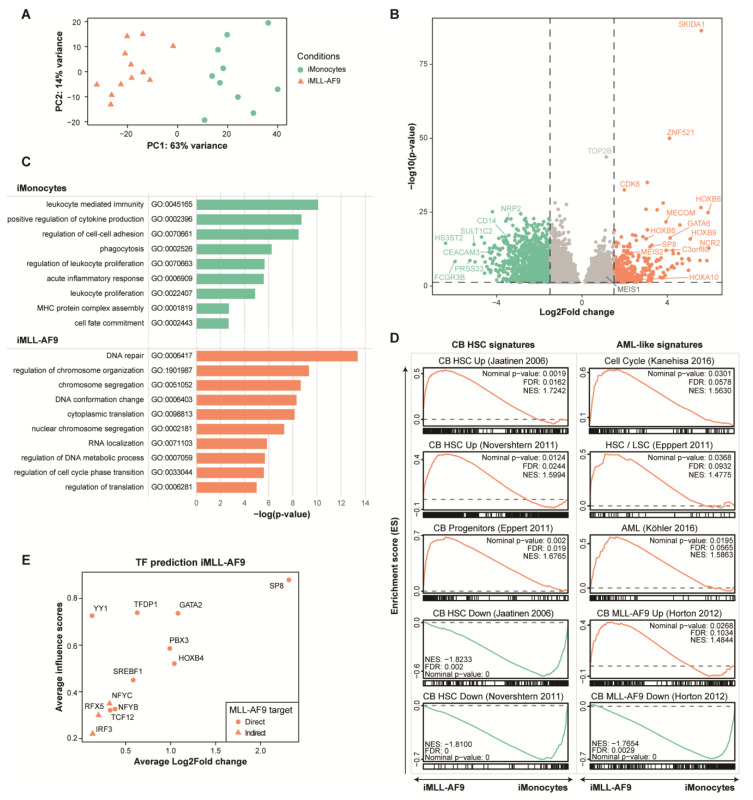
Identification of MLL-AF9-induced expression changes and key transcription factors. (**A**) PCA plot of iMonocytes and iMLL-AF9 cells. RNA-seq samples collected from three independent differentiation experiments, including at least three biological replicates each between day 20 and day 28. Shapes and colors represent the conditions. (**B**) Volcano plot representing differentially expressed genes of iMLL-AF9 cells compared to iMonocytes. Y-axis depicts -log of the *p*-values and x-axis the log2 fold change. Dotted lines signify a cut-off for significance (*p*-value < 0.05) and an absolute log2 fold change > 1.5. The color green and orange indicate significantly upregulated genes in iMonocytes and iMLL-AF9 cells, respectively. (**C**) Significantly enriched and summarized GO-terms important in either iMonocytes or iMLL-AF9 cells, including -log of *p*-values for each term. (**D**) Gene set enrichment figures with CB HSCs [33,34], CB progenitors [35], cell cycle genes [36], leukemic stem cells (LSC) [35], acute myeloid leukemia [37], and MLL-AF9 CB signatures [6]. Depicted are the enrichment scores (ES), nominal *p*-values, false discovery rates (FDR), and normalized enrichment scores (NES). The colors signify the condition in which the gene set is enriched—orange represents iMLL-AF9 cells and green represents iMonocytes. (**E**) Scatterplot representing the average log2 fold change (x-axis) and average inferred influence scores (y-axis) of TFs important in iMLL-AF9 cells compared to iMonocytes from two independent differentiation experiments. The influence score represents how well differences in two cell states can be explained by a TF. The shape represents direct binding of the MLL-AF9 fusion protein to the respective TF locus.

**Figure 3 cells-12-01195-f003:**
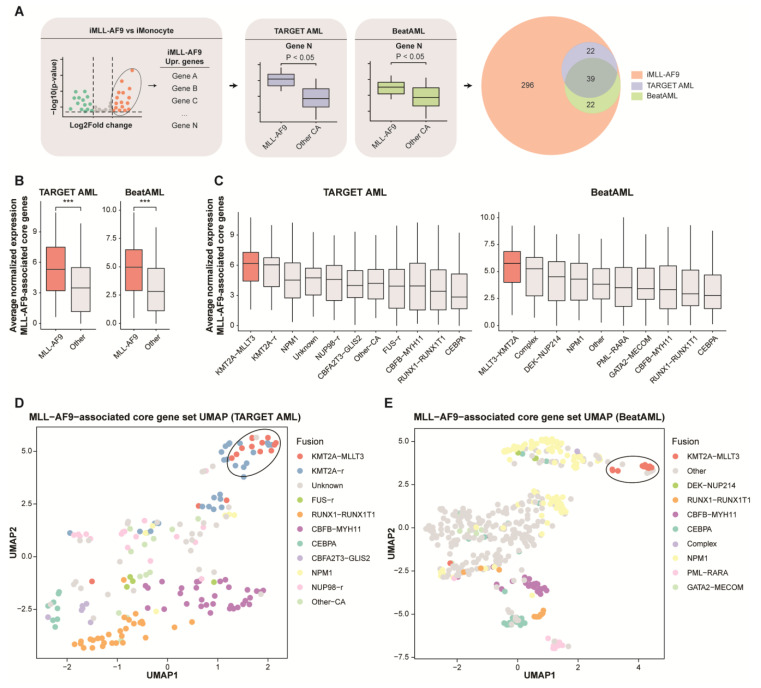
Uncovering high-confidence MLL-AF9-associated genes in primary AML. (**A**) Schematic representation of identifying MLL-AF9-associated genes. First, significant upregulated genes in iMLL-AF9 cells compared to iMonocytes are uncovered. For each upregulated gene, the expression was determined and statistically tested (one-way ANOVA) in two independent primary AML cohorts: (pediatric) TARGET AML and (adult) BeatAML. CA stands for cytogenetic abnormality. Finally, 39 genes that were significantly upregulated in both cohorts as well as the iMLL-AF9 cells, were considered MLL-AF9-associated core genes. (**B**) Boxplots depicting the mean log-normalized expression value of MLL-AF9-associated core genes (*n* = 61) for each MLL-AF9 patient versus patients with other AML subtypes, for each respective cohort (TARGET AML and BeatAML). The whiskers represent standard deviation; edges depict the inter-quartile ranges. The black center line represents the median. Three asterisks signify *p*-value < 0.001. (**C**) Boxplots illustrating the mean log-normalized expression value of the MLL-AF9-associated core genes per AML subtype in each respective cohort (TARGET AML and BeatAML). The whiskers represent standard deviation, edges depict the inter-quartile ranges, and the black center line illustrates the median. (**D**) UMAP visualization of 187 primary AML samples from the TARGET AML pediatric cohort using the expression of 39 MLL-AF9-associated core genes. Dots are colored according to cytogenetic abnormality annotation. (**E**) Similar to (**D**), except using 460 primary AML samples from the BeatAML adult cohort.

**Figure 4 cells-12-01195-f004:**
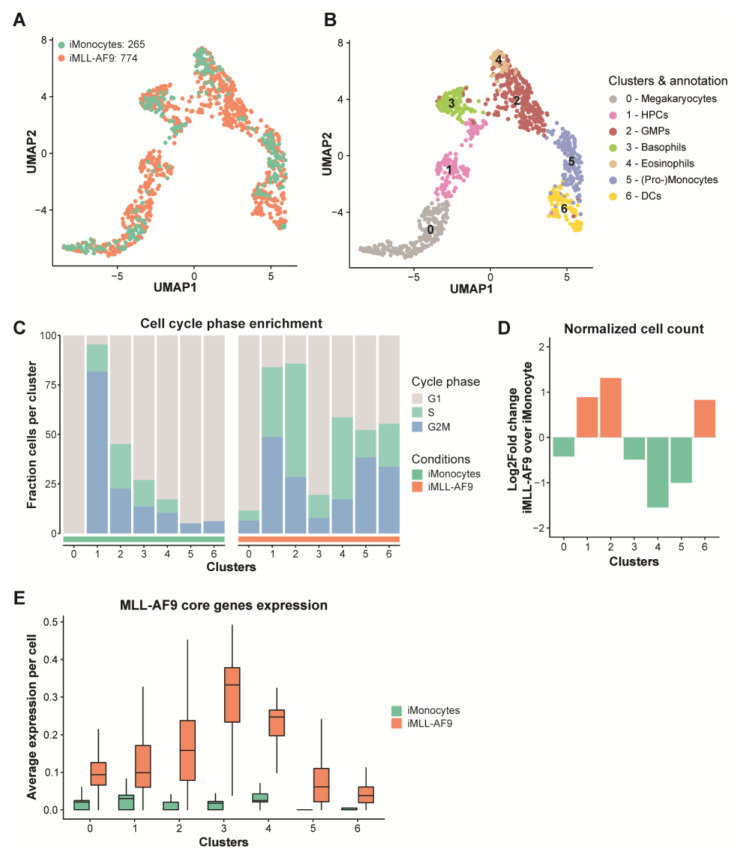
Single-cell RNA-seq analysis of iPSC-derived iMonocytes and iMLL-AF9 cells. (**A**) UMAP visualization of 265 iMonocytes and 774 iMLL-AF9 cells. Dots are colored according to conditions. (**B**) UMAP visualizing Louvain clustering. Clusters are numbered and colored. Each cluster is annotated according to expression of established markers. (**C**) Bar chart representing the fraction of cells enriched for a specific cell cycle phase (G1, G2M, S) for each cluster and condition. (**D**) Bar chart illustrating the normalized cell count fold change of iMLL-AF9 cells over iMonocytes. The orange color represents higher cell count for the MLL-AF9 induced condition and green represents higher cell count in the iMonocyte condition. (**E**) Boxplots illustrating the mean normalized and denoised expression of MLL-AF9-associated core genes for each cell per cluster. The whiskers represent standard deviation, edges depict the inter-quartile ranges, and the black center line illustrates the median. Colors represent the conditions.

## Data Availability

The datasets analyzed during the current study are available in the Gene Expression Omnibus (GEO) repository GEO Series GSE217015 and GSE220212.

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
