# Peer review of "Inducible MLL-AF9 Expression Drives an AML Program during Human Pluripotent Stem Cell-Derived Hematopoietic Differentiation"

_cells, 2023, doi:10.3390/cells12081195_

Round 1

Reviewer 1 Report

This is a well written paper describing a novel in vitro model to study how the MLL-AF9 fusion protein initiates leukemogenesis. They demonstrate that activation of MLL-AF9 in differentiating human iPSCs results in changes in the differentiation profile. Molecular characterisation (by RNA sequencing) of the cell populations generated in the presence of MLL-AF9 revealed transcriptional similarities with primary MLL-AF9+AML cells. The authors go on to characterise core genes that are regulated by MLL-AF9. Albeit with some caveats (see below), this platform provides a really useful model to study leukaemia initiation with the potential in future to identify and test new therapeutic strategies.

There are few points that could be addressed either by the addition of experimental results or by the modification of their interpretation and conclusions to take these issues into account.

1.The authors show by western blotting that the DOX induction works well in undifferentiated iPSCs. However, it is also important to show the response at the time point that the induction is done (ie at day 4 or later). This is important because there are reports that DOX inducible systems that work really well in undifferentiated iPSC do not function so efficiently as cells differentiate. This could be due to silencing, even within the AAVS1 locus that has been considered as a “safe harbour”

https://doi.org/10.12688/f1000research.19894.2

https://www.ncbi.nlm.nih.gov/pmc/articles/PMC7047106/

2.Dox is added at day 4 – this is quite early in the differentiation protocol, significantly earlier than the timing of the appearance haematopoietic progenitor cells (HPCs) in the standard differentiation protocols that are used. Can the authors state when HPCs first appear in the protocol they are using?  If day 4 is prior to the presence of HPCs, then the authors should justify why they chose that time point and acknowledge how this might affect the interpretation of their data.

3. Following DOX induction, the iMLL-AF9 cells do not express CD14 nor CD64 implying that the cells are more immature. In addition to these markers the authors could show the expression of HPC markers that are commonly used to define HPCs from human iPSCs (eg CD45, CD34, CD43, CD44) during the differentiation process (see point 2) and in the final iMLL-AF9 cells.  This is important because the context in which the fusion protein is activated will determine the downstream targets (as they rightly point out themselves).

4. For RNA sequencing the authors should state clearly in the results section when cells were harvested for analyses. The methods and the figure legend state “between days 20 and 28” but it is noted that the iMonocytes are exhausted by day 30 so the earlier timepoints would ensure the analyses of equivalent cell populations that are not exhausting? It would be informative to label samples from each day/experiment on the PCA plot to enable the reader to match the appropriate control. There are 12 iMLL-AF9 samples and only 10 iMonocyte samples so they are not all matched?

5. Given the potential transgene silencing issue mentioned above, it is unclear how homogeneous the iMLL-AF9 cell population is following DOX induction.  Is DOX active in all cells and do all cells in the population express equivalent levels of MLL-AF9?  Can the authors detect expression of AML-AF9 fusion in the single cell analyses?  If this is possible and there is heterogeneity, then only the cells that express the fusion could be selected/gated and analysed. This could eliminate some of the ‘noise’ and would likely make their data even stronger. 

6.If the iMLL-AF9 population is heterogeneous with respect to MLL-AF9 expression the authors might have to modify their conclusion that the response to the fusion is context dependent. If it isn’t possible to demonstrate the expression of the fusion in the single cell dataset then the authors should discuss this potential caveat in the discussion.

7. In the abstract, the authors state that MLL-AF9 expression results in “expansion of CD34+ HPCs”.  I think this is slightly misleading as they haven’t actually shown expansion of this cell population directly. Rather their RNA sequencing data shows that a cluster that is annotated as HPCs is increased following MLL-AF9 induction.  The authors could show flow cytometry data that support this statement more directly or a more accurate interpretation of the RNA sequencing should be stated in the abstract.

Minor

Lines 35-36: this is not a complete sentence

Line 306: When describing the GSEA data, “anti-correlated” should be ‘was inversely correlated with

Reviewer 2 Report

This is a well written manuscript using a doxycycline-inducible MLL-AF9 hiPSC model to describe the disruption of normal myelomonocytic differentiation upon MLL-AF9 expression and identification of MLL-AF9 specific gene expression profiles that are consistent with gene expression signatures of primary MLL-AF9 AML patient data.

Specific comments:

1. While the authors acknowledge the original manufacturer Sanquin of the hiPSC line MML6838.Cl2 (MIP 7 Cl.1-2) in Material and Methods - Cloning strategy, they do not cite the original paper (PMID 28395797). Moreover, the sub-title Cloning strategy is somewhat misleading as the description does not detail any cloning strategy of the MLL-AF9 fusion product into the donor vector (presumably the PP1R12C-pTREG3-CAGGS-Tet3G vector described in PMID 24497442 that the authors cite within their reference 11, which needs to be reformatting as it only provides the authors’ initials and not full last name) per se but foremost describes in brief the nucleofection of the vector(s) and gRNAs (as presumably described in PMID 23287722) into the cells and selection using puromycin. The manuscript does not provide the primers used for identification of positive clones, nor does it provide any characterization (normal karyotype, pluripotency, etc) of the derived Dox-inducible MLL-AF9 hiPSC line, which is standard for any newly derived or engineered pluripotent stem cell line.

2. The acronym “GRNs” is mentioned in Material and Methods - ANalysis Algorithm for Networks Specified by Enhancers (ANANSE), however what it stands for is not stated. I presume it stands for gene regulatory networks. Please clarify.

3. Figures are usually numbered and mentioned in order. However, Supplementary Figure S2D is mentioned first (Material and Methods - Single cell RNA sequencing), followed by Supplementary Figure S2D before Figure 1A is mentioned in Results.

4. Regarding Figure 1B, there is no description of the Western blot procedure nor any information of antibodies used (clone, manufacturer, etc). This is especially important information as the Western blot (Figure 1B and provided original blot) shows many bands, not just one for vinculin and one for the MLL-AF9 fusion product. Such “unspecific” bands are rather unusual for Western blots developed using antibodies specific to the target protein without known reasons for appearance of potential degradation products or known isoforms. Please add a description in the Material and Methods section and explain the multiple bands detected.
